# Social Media Exposure and Muscle Dysmorphia Risk in Young German Athletes: A Cross-Sectional Survey with Machine-Learning Insights Using the MDDI-1

**DOI:** 10.3390/healthcare13141695

**Published:** 2025-07-15

**Authors:** Maria Fueth, Sonja Verena Schmidt, Felix Reinkemeier, Marius Drysch, Yonca Steubing, Simon Bausen, Flemming Puscz, Marcus Lehnhardt, Christoph Wallner

**Affiliations:** Department for Plastic and Hand Surgery, BG University Hospital Bergmannsheil, Ruhr University Bochum, Bürkle-de-la-Camp Platz 1, 44789 Bochum, Germany

**Keywords:** body muscle dysmorphia, body image, body image disorder, training, social media

## Abstract

**Background and Objectives:** Excessive social media use is repeatedly linked to negative body image outcomes, yet its association with muscle dysmorphia, especially in athletic youth, remains underexplored. We investigated how social media exposure, comparison behavior, and platform engagement relate to muscle dysmorphia symptomatology in young German athletes. **Materials and Methods:** An anonymous, web-based cross-sectional survey was conducted (July–October 2024) of 540 individuals (45% female; mean age = 24.6 ± 5.3 years; 79% ≥ 3 h sport/week) recruited via Instagram. The questionnaire comprised demographics, sport type, detailed social media usage metrics, and the validated German Muscle Dysmorphic Disorder Inventory (MDDI-1, 15 items). Correlations (Spearman’s ρ, Kendall’s τ) were calculated; multivariate importance was probed with classification-and-regression trees and CatBoost gradient boosting, interpreted via SHAP values. **Results:** Median daily social media time was 76 min (IQR 55–110). Participants who spent ≥ 60 min per day on social media showed higher MDDI scores (mean 38 ± 7 vs. 35 ± 6; *p* = 0.010). The strongest bivariate link emerged between perceived social media-induced body dissatisfaction and felt pressure to attain a specific body composition (Spearman ρ = 0.748, Kendall τ = 0.672, *p* < 0.001). A CatBoost gradient-boosting model out-performed linear regression in predicting elevated MDDI. The three most influential features (via SHAP values) were daily social media time, frequency of comparison with fitness influencers, and frequency of “likes”-seeking behavior. **Conclusions:** Intensive social media exposure substantially heightens muscle dysmorphia risk in young German athletes. Machine-learning interpretation corroborates time on social media and influencer comparisons as primary drivers. Interventions should combine social media literacy training with sport-specific psychoeducation to mitigate maladaptive comparison cycles and prevent downstream eating disorder pathology. Longitudinal research is warranted to clarify causal pathways and to test targeted digital media interventions.

## 1. Introduction

Body image formation in emerging adulthood is increasingly shaped by digital media environments. Platforms such as Instagram, TikTok, and Facebook deliver an uninterrupted stream of highly curated physique ideals and thereby intensify upward social comparison, which is consistently associated with body dissatisfaction, disordered eating, and depressive symptomatology [1,2,3,4]. Within the spectrum of appearance-related disturbances, muscle dysmorphia (MD) constitutes a distinct subtype characterized by a persistent belief of insufficient muscularity despite objectively adequate muscle mass [5,6]. Meta-analytic evidence indicates elevated MD symptomatology among competitive bodybuilders compared with non-bodybuilder resistance trainers [7,8]; however, data on endurance and hybrid athletes remain scarce.

From a theoretical perspective, these developments can be understood through the lens of objectification theory, which posits that individuals, particularly those in visually driven, appearance-focused environments, internalize an observer’s perspective on their own body [9]. This self-objectification might lead to heightened self-surveillance, chronic body monitoring, and vulnerability to appearance-based anxiety. Social media platforms provide both the setting and reinforcement mechanisms (e.g., likes, views, and comments) that make these processes especially salient. Moreover, the tripartite influence model suggests that peers, media, and parents influence body image through appearance comparison and internalization of societal ideals, two mechanisms highly relevant in the context of social media use and athletic identity [10].

Recent research has further highlighted how specific Instagram-based activities, such as browsing appearance-focused images, posting selfies, and following beauty or fitness influencers, correlates to higher body dissatisfaction and acceptance of cosmetic surgery among young women [11]. 

Social media (SM) use appears to act as an amplifier of muscular and leanness ideals [12]. Frequency of platform engagement, intensity of physique-focused comparison, and reliance on positive feedback cues (“likes”) correlate with more severe body image pathology in adolescents and young adults in a dose-response fashion [2,3,13]. Yet no large-scale study has quantified how specific SM behaviors relate to MD risk in German athletic populations.

The Muscle Dysmorphic Disorder Inventory-1 (MDDI-1) is a psychometrically robust, gender-invariant instrument for assessing MD symptom severity in German-speaking samples [14]. To date, it has seldom been applied outside bodybuilding contexts. Combining conventional correlation techniques with explainable machine learning (e.g., classification-and-regression trees and CatBoost with SHAP interpretation) offers a data-driven means of ranking the SM features most predictive of elevated MDDI scores, an approach largely absent from previous research.

### State of Research and Study Objective

Recent empirical studies have begun to explore how specific social media behaviors relate to body image disturbances, particularly in the context of muscularity-oriented ideals (Table 1). For instance, Schoenenberg and Martin (2020) demonstrated that exposure to fitness-related imagery on Instagram is linked to internalization of male body ideals and higher muscle dysmorphia scores, though their work was limited to male users and did not apply machine learning [15]. Similarly, Imperatori et al. (2022) found that muscle dysmorphia symptoms mediate the relationship between social media addiction and disordered eating behaviors in young adults, underscoring the role of digital environments as risk amplifiers [16]. Giordano et al. (2025) emphasized the moderating role of social support and self-efficacy, while Schneider et al. (2017) provided experimental evidence that muscle dysmorphia can affect social cognition and behavior [17,18]. Furthermore, Wallner et al. (2022; 2023) highlighted how digitally mediated beauty ideals influence aesthetic self-perception and body satisfaction, particularly in women [19,20]. While these studies have significantly advanced the field, they are often limited to gender-specific populations, non-athletes, or single-method approaches. The present study aims to build on and integrate this body of work by examining a large-scale investigation to combine a sport-diverse, mixed-gender German athlete sample with interpretable machine learning techniques (CatBoost + SHAP) and a validated German version of the Muscle Dysmorphic Disorder Inventory (MDDI-1). Unlike earlier work that often focused on male bodybuilders, non-athletic populations, or bivariate associations alone, our approach allows for a nuanced, data-driven prioritization of digital behavioral risk factors. This integrative design not only strengthens external validity but also offers practical implications for targeted interventions in athletic and clinical settings.

Accordingly, the present cross-sectional study aimed to
Determine the prevalence of clinically relevant MD symptoms in a nationwide sample of young German athletes engaged in strength, endurance, or hybrid training;Quantify the association between SM exposure (time and engagement metrics) and MDDI-1 scores;Identify via interpretable machine-learning models the digital predictors that most strongly increase MD risk.

## 2. Materials and Methods

### 2.1. Study Design

A web-based survey, encompassing multiple sub-sections, was developed to explore the predominant aspects associated with muscle dysmorphia. The initial segment solicited general demographic information, age of onset for sports participation, the specific sport engaged in, and the duration of sports activity. Subsequently, the questionnaire integrated the comprehensive elements of the validated German Muscle Dysmorphic Disorder Inventory (MDDI-1) alongside a custom-designed set of items assessing the temporal and content-related use of social media as well as its influence on self-perception, appearance comparison, and consumption behavior. These social media-related questions were theoretically grounded in objectification theory and the tripartite influence model, capturing key mechanisms such as self-surveillance, internalization of ideals, and peer/media comparison [21]. Each item was analyzed individually rather than as composite scores, allowing for precise, behavior-level analysis and integration with the machine learning model.

This survey was administered through the SurveyMonkey platform and promoted via Instagram; we included German participants between the ages of 18 and 40 (SurveyMonkey Inc., San Mateo, CA, USA) [14], 55% of which identified as male, 45% as female, and 1% as diverse. Furthermore, 36% of respondents identified themselves as being from cities with populations of less than 100,000, 40% from cities with populations of 100,000–1,000,000, and 24% from cities with populations of more than 1,000,000. Most of those surveyed were aged between 25 and 29 years old (Figure 1).

### 2.2. IP Classification and Verification of Origin 

Within the framework of the SurveyMonkey platform, the IP address of each respondent was anonymously recorded through IP tracking. This process was employed to ascertain the geographic origin of participants and to exclude bots and fraudulent IP addresses. To achieve this, the IP addresses provided by SurveyMonkey were correlated with their actual locations using the Geolocation by IP address, which accessed the application programming interfaces (APIs) of the website https://geolocation-db.com (accessed on 20 December 2023) on as described before. The location heatmap was generated using Folium 0.15.1 (Figure 1).

### 2.3. Operationalization of Social Media Variables

Social media behavior was assessed using a custom social behavior questionnaire, which included the following items:Daily time spent on social media (five ordinal categories from “none” to “>120 min”);Perceived influence of social media on body image (Likert scale 1–5);Proportion of social media content related to fitness/sports (6-category scale);Degree of preoccupation with nutrition (6-category scale from “not at all” to “weigh and track everything”);Frequency of prioritizing sport over social obligations (Likert scale 1–6);Discomfort when eating spontaneously (Likert scale: “not acceptable” to “always acceptable”);Ability to accept body-related compliments (Likert scale 1–5);Visual ideal body selection (image-based response);Attention to others’ body appearance (Likert scale 1–6);History of bulking/cutting phases (yes/no).

Items targeting shared constructs demonstrated acceptable internal consistency (Cronbach’s α = 0.76). However, given the novelty of this instrument, we acknowledge the need for further validation, as noted in the Limitations Section.

### 2.4. Identification of Correlation and Classification and Regression Tree (CART) Analysis

Initial associations between social media variables and MDDI-1 scores were examined using Spearman’s ρ and Kendall’s τ, suitable for ordinal data and non-normal distributions. In our study, multidimensional scaling (MDS) was employed to visualize the high-dimensional relationships in our dataset, reducing it to two dimensions for clarity (see Appendix A). This method, alongside a correlation heatmap, offered insights into the intricate connections between variables related to demographics, muscle training, body satisfaction, and social media activities. Spearman’s rho and Kendall’s tau tests further quantified these relationships, providing robust statistical measures of correlation strengths and directions. To investigate significant correlations within the dataset, a machine learning algorithm was employed, specifically CART (classification-and-regression tree) analysis, a technique of binary recursive partitioning that is well established in the field. This method involves the selection of various measures of purity, referred to as splitting functions, with the Gini index being utilized to quantify the entropy present within the cohorts. Furthermore, correlation attribute evaluation was applied to assess the significance of features. The configuration for the decision tree classifier, implemented using version scikit-learn 1.4, included a training set comprising 80% of the data and a testing set constituting the remaining 20%. To enhance the model’s performance, hyperparameter tuning was undertaken. Additionally, 5-fold cross-validation methods were executed to ascertain the model’s efficacy, and scores denoting the importance of features were computed.

### 2.5. Employing CatBoost and SHAP Values for Enhanced Predictive Analysis and Model Interpretability

To identify key behavioral predictors of muscle dysmorphia symptoms, we applied an interpretable machine learning approach using CatBoost 1.2.3, a gradient boosting algorithm optimized for structured data. CatBoost is particularly well suited for this research context, as it processes categorical variables natively, reduces the risk of overfitting in medium-sized datasets, and performs robustly even in the presence of multicollinearity and non-linear relationships—conditions under which traditional regression techniques often fall short.

Hyperparameters were tuned using a structured grid search and 5-fold cross-validation on the training set. The final model was trained with 1000 iterations, a tree depth of 8, and a learning rate of 0.1. Model performance was evaluated on the hold-out test set using standard metrics: root mean squared error (RMSE): 4.88 and coefficient of determination (R^2^): 0.49.

For interpretability, we used SHapley Additive exPlanations (SHAP), which provide insight into how individual features contribute to the model’s predictions. SHAP values are derived from game theory and allow for transparent, case-level feature attribution, which is especially important in health-related predictive modeling.

In line with the goals of this study, this interpretable machine learning approach was selected not merely for predictive power but as a conceptually aligned tool to prioritize social media-related risk factors in a data-driven, explainable manner. Visualizations of the model output were created using Adobe Illustrator (v28.3; Adobe Inc., Salt Lake City, UT, USA) for clarity and publication-quality rendering.

## 3. Results

### 3.1. Survey Population

A total of 540 participants from Germany completed the survey through the platform SurveyMonkey. Of those, 241 participants were female (45%), 295 male (55%), and 3 of other gender (<1%). Furthermore, 77 participants were younger than 20 (14%), 144 between 20 and 24 years (27%), 182 between 25 and 29 (34%), and 136 participants 30 years or older (25%). In addition, 196 participants (36%) lived in a city with fewer than 100,000 inhabitants, 218 (40%) in a city with more than 100,000 inhabitants and less than 1,000,000, and 126 (24%) lived in a city with more than 1,000,000 inhabitants. The demographic data of the participants are visualized in Figure 1. Of these participants, 425 (79%) exercised for more than 3 h per week, and 58% of all participants exercised for more than 6 h per week. The survey revealed that 31% of respondents primarily engage in weight training, while 52% of respondents primarily engage in endurance sports. A further 16% described their sport as a hybrid of strength and endurance training or, alternatively, as a combination of both. Forty-five respondents indicated that they engage in another athletic activity.

### 3.2. Identification of Correlations

The correlation heatmap revealed significant linkages among variables, notably those relating to social media’s influence on body image, eating behavior, and comparisons with fitness influencers, showcasing strong correlations. The key findings with the highest correlation were the influence of social media on body dissatisfaction and body composition pressure and the comparison to influencers and perception of body image (Figure 2). Appendix A includes the heatmap for detailed examination. Subsequently, correlations with indices above 0.5 were rigorously analyzed using Spearman’s rho and Kendall’s tau tests to assess categorical correlations, providing a statistical foundation to understand the nuanced impacts of social media on individuals’ perceptions and behaviors related to physical fitness and body image.

#### 3.2.1. The Use of Social Media Has Been Linked to Increased Pressure to Conform to a Specific Body Ideal, Which Is Often Portrayed as the Norm on These Platforms

The data reveal notable correlations, with the highest between the item “The influence of social media has made me more dissatisfied with my own body” and “I feel pressured by social media to have a certain body composition”, demonstrating Spearman’s rho at 0.7480 and Kendall’s tau at 0.6720. These values indicate a robust positive relationship. Similarly, the items “I compare myself to influencers (Instagram/TikTok/YouTube)” and “Social media influences my perception of my own body image” also demonstrate robust positive correlations (Spearman’s rho: 0.7117; Kendall’s tau: 0.6318). These findings are corroborated by highly significant *p*-values, which underscore the robust associations between social media’s negative impacts and pressures on body image, as illustrated in Figure 2. Furthermore, a strong positive correlation was identified between the statements “The influence of social media has made me more dissatisfied with my own body” and “Social media influences the perception of my own body image”. This correlation was quantified by Spearman’s rho at 0.6811 and Kendall’s tau at 0.6043 in Figure 2.

#### 3.2.2. The Influence of Social Media on Eating Behavior May Contribute to an Increased Prevalence of Eating Disorders

Regarding eating habits, there is a strong positive correlation between the statement “Social media/influencers influence my eating habits in everyday life” and “The influence of social media has made me more dissatisfied with my own body”, as evidenced by a Spearman’s rho of 0.56. The results also indicate a significant positive correlation between the aforementioned eating behavior and the statement “I feel pressured by social media to have a certain body composition” as well as the following variables: Kendall’s tau of 0.5175 and Spearman’s rho of 0.5715.

#### 3.2.3. Social Media Has a Significant Influence on Our Body Perception

The results of our survey indicate a significant correlation between the impact of social media on one’s body image and the pressure individuals experience to attain a specific body composition that is commonly portrayed on social media platforms. This relationship is evidenced by a Spearman’s rho correlation coefficient of 0.6246 and a Kendall’s tau value of 0.5515. Furthermore, this correlates with an increasing dissatisfaction with one’s own body through the consumption of social media and the comparison to influencers on Instagram/TikTok and YouTube, as shown by Spearman’s rho at 0.5953/0.5947 and Kendall’s tau at 0.5454/0.5175 (Figure 2). Additionally, 60.42% of the respondents indicated that social media had an impact on their body image, with 46.02% perceiving social media as a source of body dissatisfaction and 46.02% reporting that social media influenced their dietary supplement consumption.

#### 3.2.4. Social Media Time Deteriorate MDDI Scores

The Kruskal–Wallis H-test was utilized to determine differences in MDDI scores across three ordinal categories of social media usage time. The MDDI score is 38 points for respondents who engage in social media for a minimum of 60 min per day and 35 points for those who engage in social media for a minimum of 30 min per day. Social media platforms are saturated with images of highly muscular and aesthetically ideal bodies, often showcased by fitness influencers and bodybuilders. Constant exposure to these idealized body standards can lead individuals to feel inadequate about their own muscularity, contributing to higher MDDI scores (Figure 3). The significant result (*p* = 0.0103) indicated a correlation between the amount of time spent on social media and higher scores on the MDDI (see Figure 3). The relationship between time spent on social media and higher MDDI scores can be elucidated by examining several interrelated factors. For instance, many participants indicated that social media influenced their body image, with 46.02% perceiving it as a source of body dissatisfaction.

## 4. Discussion

This study provides robust evidence that social media engagement is intricately linked to muscle dysmorphia risk among young German athletes. Three findings stand out. First, social media induced body dissatisfaction was strongly correlated with perceived pressure to attain a specific physique, supporting the premise that appearance-focused content acts as a powerful external stressor on self-evaluation. Second, particularly comparison behavior with fitness influencers emerged as the strongest behavioral correlate of heightened MDDI scores, echoing prior work on upward social comparison and body image disturbance [1,2]. Third, machine learning interpretation identified social media time, influencer comparison frequency, and reinforcement-seeking behavior (e.g., seeking “likes”) as the most influential variables associated with elevated MD symptoms, extending earlier correlational studies by quantifying the relative weight of each digital exposure.

### 4.1. Integration with the Existing Literature

Our findings align with international data linking high social media use to body image concerns and eating disorder pathology [3,13] but add nuance by focusing on muscularity-oriented ideals within a mixed-gender athletic cohort. Previous German studies, such as that by Schoenenberg and Martin (2020), have investigated this phenomenon exclusively in male participants [15]. The observed gender pattern of greater dissatisfaction with body fat in women and concerns of insufficient muscularity in men mirrors dual-pathway models that differentiate thinness and muscularity ideals [4]. These findings are consistent with previous research from our department, which has shown that digital media significantly shape aesthetic standards and contribute to body-related pressures, particularly in the context of female breast perception and social comparison mechanisms. Our current study extends this body of work by focusing on muscularity-oriented body ideals in athletes and quantifying digital risk factors for muscle dysmorphia using interpretable machine learning [19,20].

The high correlations between social media pressure, peer comparison, and diet modification further support cognitive–behavioral frameworks in which external appearance standards might trigger maladaptive coping behaviors, including restrictive eating and supplement overuse [7]. In relation to the existing literature, Giordano et al. (2025) emphasized the moderating influence of social support and self-efficacy on the development of MD symptoms. Our findings build on this by highlighting platform-specific behavioral drivers, such as time spent on social media, comparison with influencers, and validation-seeking, and demonstrating their predictive relevance through interpretable machine learning methods [17].

Similarly, Imperatori et al. (2022) identified muscle dysmorphia as a mediating factor between social media addiction and disordered eating. While our results align with this pathway, we extend the discussion by focusing on athletes specifically and highlighting muscularity-focused comparison behaviors, offering a targeted lens on this psychosocial mechanism [16]. Finally, Schneider et al. (2017) provided experimental evidence that MD and body schema priming can reduce social interaction desire. We advance this line of research by analyzing naturally occurring digital behaviors in a large, real-world sample, illustrating how frequent influencer comparison and media-induced body pressure may contribute to the internalization of maladaptive body-related schemas [18].

### 4.2. Mechanistic Considerations

Social comparison theory and objectification perspectives offer plausible mechanisms: Athletes who are exposed to idealized, digitally edited physiques repeatedly compare themselves to unrealistic standards, internalize those ideals, and experience heightened self-surveillance, which is further amplified by instantaneous feedback mechanisms such as likes and follower counts. The resulting dissatisfaction appears to function as a central hub linking digital exposure to downstream behavioral changes, including altered eating habits, as illustrated in our network of correlations [22].

### 4.3. Practical Implications

Given that more than half of our sample exceeded one hour of daily social media use and that ≥60 min/day was associated with statistically significant and potentially elevated MDDI scores, routine screening for social media related body image stressors should be incorporated into sports medicine and university health services. Media literacy programs that target comparison behavior and promote body appreciation may mitigate risk, while coaches and fitness influencers could model healthier content practices.

### 4.4. Strengths and Limitations

The strengths of the current study include the large, sport-diverse sample; use of a validated MD measure (MDDI-1); and complementary machine learning analysis that enhanced variable prioritization. Limitations are inherent to the cross-sectional, self-report design. Causal inferences cannot be drawn, as survey promotion via Instagram may have attracted heavier social media users, and unmeasured confounders (e.g., baseline psychopathology) could bias associations. Potential confounders such as baseline body image dissatisfaction, perfectionistic traits, or general psychopathology were not assessed in this study and may have influenced both social media engagement and MD symptomatology. Nevertheless, the convergence of traditional and ML results bolsters confidence in the identified risk network.

### 4.5. Future Directions

Prospective studies should monitor athletes longitudinally to clarify temporal ordering between SM exposure and MD onset, test causal mechanisms experimentally (e.g., SM abstinence or content-manipulation trials), and evaluate tailored interventions. Incorporating objective SM-usage metrics and physiological outcomes (e.g., injury rates from excessive training) would provide a more comprehensive risk profile.

## 5. Conclusions

Intensive social media engagement, especially the comparison behavior with fitness influencers, is strongly associated with elevated muscle dysmorphia symptoms and body dissatisfaction in young German athletes. Appearance-related pressure and socially reinforced comparison behaviors form a tightly interconnected risk network that also extends to eating-related behaviors. These findings highlight the importance of integrating targeted media literacy and body appreciation strategies into athletic settings to reduce vulnerability to appearance-based stressors.

While the present study provides important insights, certain methodological considerations should be noted. The cross-sectional design allows for the identification of associations but does not permit causal conclusions. Self-reported social media usage may be influenced by estimation bias, although such measures remain common in psychosocial research. Additionally, while our recruitment via Instagram ensured access to an ecologically valid and relevant population, it may have resulted in an overrepresentation of more digitally engaged individuals. Despite these considerations, the study offers a valuable contribution through its use of a large, sport-diverse sample; a validated MD instrument (MDDI-1); and interpretable machine learning techniques that help identify key behavioral risk factors. Future longitudinal and experimental research should aim to clarify causal relationships and evaluate the impact of targeted prevention strategies in both digital and athletic environments.

## Figures and Tables

**Figure 1 healthcare-13-01695-f001:**
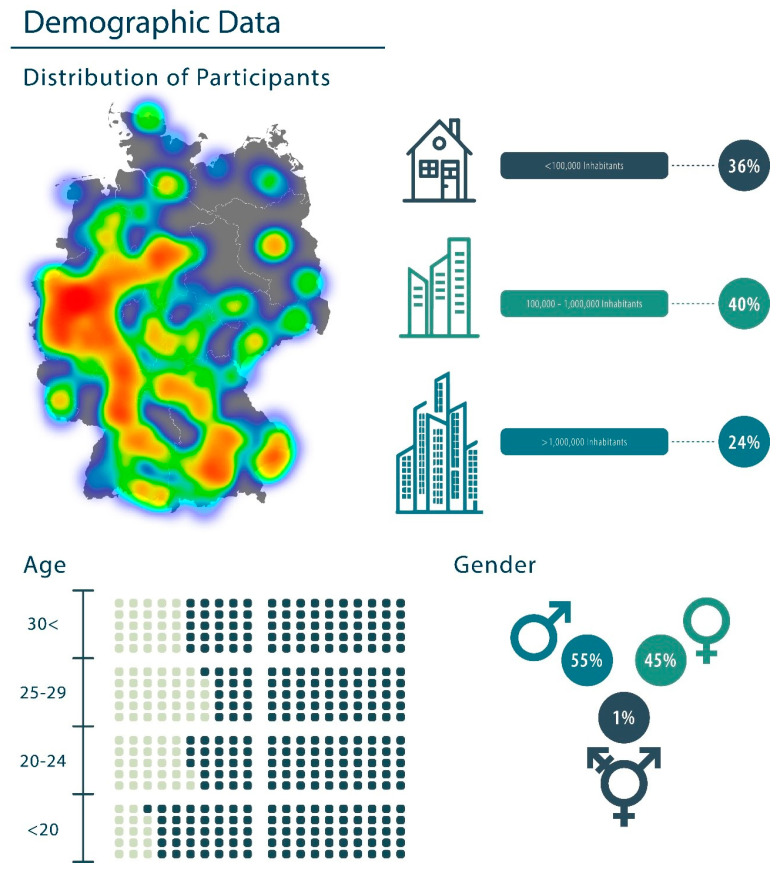
Geographical distribution of the study participants: The heatmap depicts the geographical distribution of the study participants.

**Figure 2 healthcare-13-01695-f002:**
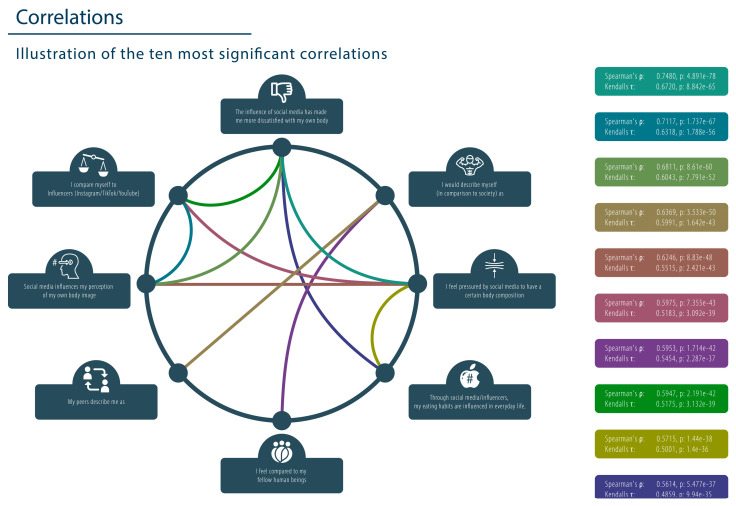
This circular correlation plot visualizes the ten strongest associations between self-reported social media behaviors and body image-related statements from the social behavior questionnaire. Each node represents a specific item, while connecting lines indicate statistically significant correlations. Line colors correspond to the strength and direction of each correlation, with Spearman’s ρ and Kendall’s τ values listed in the legend to the right.

**Figure 3 healthcare-13-01695-f003:**
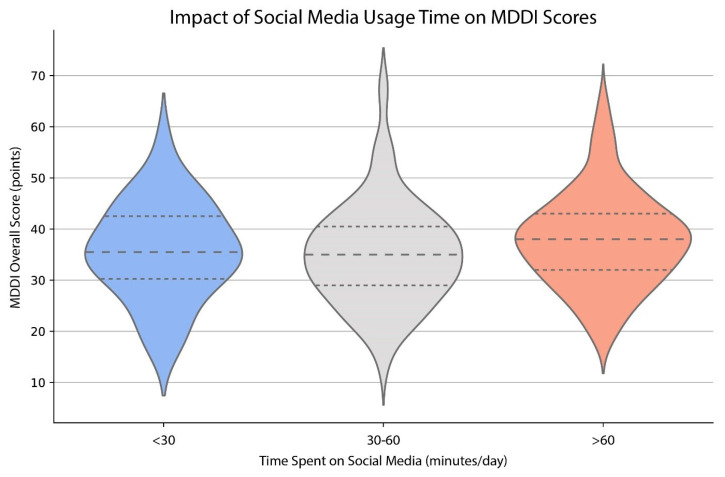
Correlation of social media usage and body muscle dysmorphia symptoms: Time spent on social media platforms correlates with a higher MDDI score. The width of each violin indicates the density of data points at different score levels. Dashed lines represent the median (center) and interquartile range (top and bottom lines). Participants who reported spending more than 60 min per day on social media exhibited higher overall MDDI scores on average, suggesting a positive association between increased screen time and muscle dysmorphia symptom severity. Kruskal–Wallis H statistic (H = 6.62, *p* = 0.0103).

**Table 1 healthcare-13-01695-t001:** Comparison of Selected Studies on Social Media and Muscle Dysmorphia.

Study	Sample	Focus	Instrument(s)	Method	Main Findings
Schoenenberg & Martin (2020) [15]	N = 203, German males	Instagram use and MD symptoms	MDDI (German)	Correlational	Instagram fitspiration linked to MD; limited to male bodybuilders
Imperatori et al. (2022) [16]	N = 721, Italian young adults (non-athletic sample)	SM addiction, eating pathology, and MD symptoms	SMAS, EDI-2, MDDI	Mediation analysis	MD mediates link between SM addiction and disordered eating; not focused athletic samples
Giordano et al. (2025) [17]	N = 2325, Italian students (non-athletic sample M/F)	Social support, self-efficacy, and MD	Custom scales + MDDI	Moderation models	Support buffers MD symptoms; SM time relevant; not focused athletic samples
Schneider et al. (2017) [18]	N = 3149, high school students (non-athletic sample M/F)	MD, body schema, and social interaction	DSM-IV Priming task + self-report	Experimental	MD priming reduces social motivation; not focused athletic samples
Present study	N = 540, German athletes (M/F)	SM behavior and MD in mixed-sport context	MDDI-1, custom SM items	Correlations + ML (CatBoost + SHAP)	Influencer comparison and SM time predict MD; novel use of explainable ML and MD symptoms in athletic individuals

## Data Availability

The datasets and materials used and/or analyzed during the current study are available from the corresponding author on reasonable request. All relevant data supporting the findings of this study are included within the article and its Appendix A, where applicable.

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
