# Peer review of "Social Media Exposure and Muscle Dysmorphia Risk in Young German Athletes: A Cross-Sectional Survey with Machine-Learning Insights Using the MDDI-1"

_healthcare, 2025, doi:10.3390/healthcare13141695_

Round 1
Reviewer 1 Report
Comments and Suggestions for Authors
In this manuscript, the authors investigated the relationship between social media exposure, comparison behavior, and platform engagement and muscle dysmorphia symptomatology in young German athletes.
This paper needs to be improved in the following aspects:
- The authors need to discuss the differences with other research works published where a similar topic is addressed. For example, the following papers cover a similar topic:
- Schoenenberg, K., & Martin, A. (2020). Relevance of Instagram and fitspiration images for muscle dysmorphia: internalization of the male beauty ideal through social media. Psychotherapeut, 65, 93-100.
- Giordano, F., Calaresi, D., Castellani, L., Verrastro, V., Feraco, T., & Saladino, V. (2025). Interaction Between Social Support and Muscle Dysmorphia: The Role of Self-Efficacy and Social Media Use. Behavioral Sciences, 15(2), 122.
- Imperatori, C., Panno, A., Carbone, G. A., Corazza, O., Taddei, I., Bernabei, L., ... & Bersani, F. S. (2022). The association between social media addiction and eating disturbances is mediated by muscle dysmorphia-related symptoms: a cross-sectional study in a sample of young adults. Eating and Weight Disorders-Studies on Anorexia, Bulimia and Obesity, 27(3), 1131-1140.
- Schneider, C., Agthe, M., Yanagida, T., Voracek, M., & Hennig-Fast, K. (2017). Effects of muscle dysmorphia, social comparisons and body schema priming on desire for social interaction: an experimental approach. BMC psychology, 5, 1-9.
- The authors do not define what research questions they try to answer. What is the main contribution in comparison with the literature?
- The paper lacks a Related Work section. This section is very important to clarify the differences (advantages and disadvantages) of this work with the literature. Authors need to include a comparative table.
- The authors used the CatBoost algorithm for predictive analysis. What is the justification for using this ensemble learning algorithm? In the literature, there are other algorithms with better results, such as Gradient Boost and AdaBoost.
- The captions of Figure 1, Figure 2, and Figure 3 contain too much text. It is necessary to reduce the description of each figure. Even this happens with the supplementary figures.
- The quality of figures must be improved. Figure 1, figure 2 and supplementary figure 2 can not be visualized correctly.
- The conclusion is shallow and lacks a critical reflection on the survey's limitations. It does not adequately address the potential challenges of analyzing social media exposure and muscle dysmorphia risk in athletes.
- The number of references is minimal for a research paper. The authors need to increase and improve the references, preferably from reputable journals.
I believe that addressing these points would make the manuscript more focused, accessible, and impactful. The improvements suggested could enhance the clarity of the research, connect it more directly to the relevant literature, and provide a more critical and comprehensive discussion of the findings and their implications.
Reviewer 2 Report
Comments and Suggestions for Authors
I have read the manuscript carefully and found that it addresses a timely and relevant issue—namely, the relationship between social media use and muscle dysmorphia symptoms in young athletes—through a combination of classical correlation analyses and interpretable machine learning techniques. This dual approach is commendable and adds value to the existing body of literature.
However, there are several areas in which the manuscript could be improved, particularly with regard to the clarity of methodological descriptions and the interpretation of findings. My detailed comments are organized below, following the journal’s review criteria.
I hope these constructive suggestions will assist both the authors and the editorial board in enhancing the quality and clarity of the work.

Reviewer 3 Report
Comments and Suggestions for Authors
Reviewer comments for authors – Major Revisions
Thank you for the opportunity to review the manuscript titled “Social-Media Exposure and Muscle-Dysmorphia Risk in Young German Athletes: A Cross-Sectional Survey with Machine-Learning Insights using the MDDI-1”. This study addresses a highly relevant topic at the intersection of social media use, body image, and mental health in athletic populations. The use of explainable machine learning (CatBoost, SHAP) alongside correlational statistics is innovative and potentially impactful.
That said, I believe the manuscript requires major revisions before it can be considered for publication. The current version would benefit from improvements in conceptual clarity, methodological reporting, statistical transparency, and engagement with the broader experimental literature.
To begin with, the introduction lays a useful foundation but would benefit from a more coherent theoretical integration. While muscle dysmorphia and body dissatisfaction are defined, the underlying psychological mechanisms, such as those proposed in objectification theory or the Tripartite Influence Model, are not fully developed. The paper would be strengthened by explicitly situating the hypotheses within such frameworks, particularly with regard to the processes of appearance comparison, internalization of body ideals, and self-surveillance. Moreover, the rationale for applying machine learning in this context should be introduced earlier and more clearly, currently, it appears as a methodological feature rather than a theoretically driven analytic choice.
The methodological section also requires further elaboration. The operationalization of key constructs (e.g., social media engagement, comparison behavior, like-seeking) is not sufficiently detailed. Were these items based on validated measures, or were they developed specifically for this study? If new, their psychometric properties (e.g., internal consistency, dimensionality) should be briefly reported or at least acknowledged as a limitation. Furthermore, the sample was recruited exclusively through Instagram, which may have introduced a significant selection bias. This point is mentioned in passing but deserves deeper discussion, especially given that the primary predictor is social media behavior.
The statistical analyses section, particularly the machine learning component, is intriguing but lacks transparency. While CatBoost and SHAP values are useful tools for prediction and interpretation, the paper does not provide sufficient detail on how the models were tuned or validated. For instance, was k-fold cross-validation employed? How were hyperparameters selected? What were the final R² and RMSE values on the test set? Without this information, it is difficult to assess the reliability of the models or compare their performance to simpler approaches (e.g., multiple regression). A comparison with a baseline model would be informative.
In the discussion, the authors rightly emphasize the central role of influencer comparison and time spent on social media, but at times the language veers into causal interpretation. Given the cross-sectional design, such interpretations should be avoided or carefully qualified. The claim that ≥60 minutes per day of social media use is associated with “clinically meaningful” muscle dysmorphia symptoms should be backed by established thresholds, or the language should be tempered. Likewise, practical recommendations (e.g., media literacy, psychoeducation) are reasonable but speculative based on the current data.
The conclusions would benefit from more cautious framing. While the results are consistent with existing literature on body image and social media, the cross-sectional nature of the design limits the strength of any inferences. In addition, important confounding variables, such as pre-existing body image concerns, perfectionism, or psychopathology, were not controlled for, and this should be acknowledged explicitly in the limitations.
I also recommend that the authors strengthen the manuscript by citing relevant experimental studies that support the proposed mechanisms linking social media exposure to body dissatisfaction and dysmorphic concerns. In particular, one recent experimental study should be considered: Di Gesto et al. (2022), who examined the effects of Instagram likes and disclaimer labels on self-awareness and body dissatisfaction among young women (https://doi.org/10.1007/s12144-021-02675-7).
These works offer empirical support for the idea that specific types of social media content and engagement can causally influence appearance-related concerns. Including them would not only enrich the theoretical background but also help position the current findings within a broader literature that includes both correlational and experimental evidence.
Lastly, some practical suggestions: the figures—particularly the heatmaps and SHAP plots—require higher resolution and clearer legends to be fully interpretable. Figure 2 could benefit from better axis labeling and visual organization. The abstract is generally clear but could be strengthened by including specific statistical findings. Finally, the English is adequate but would benefit from minor polishing to improve clarity and sentence flow, particularly in the Results and Methods sections.
Overall, this is a promising and well-conceived study on a topic of considerable social and clinical relevance. With greater theoretical grounding, improved methodological reporting, and more measured interpretation of results, this manuscript could make a strong contribution to the literature on body image and digital media. I recommend major revisions and look forward to evaluating a revised version.
Round 2
Reviewer 1 Report
Comments and Suggestions for Authors
The authors have addressed the majority of the comments in the previous round of revision. However, there are still some aspects of improvement:
- The paper lacks a Related Work section. This section is very important to clarify the differences (advantages and disadvantages) of this work with the literature. Authors need to include a comparative table. By adding this section, the number of references is increased and another suggestion is addressed.
- The authors must include in the manuscript the justification for using the CatBoost algorithm for predictive analysis instead of using other ensemble learning algorithms.
- The quality of figure 2 was not improved. This figure can not be visualized correctly.
I believe that the first two points from my comments are very important to enhance the clarity of the research, connect it more directly to the relevant literature, and provide a more critical and comprehensive discussion of the findings and their implications.
Reviewer 2 Report
Comments and Suggestions for Authors
Congratulations on the manuscript. The corrections I requested to be made have been completed.
Author Response
Thank you for your valuable review!
Reviewer 3 Report
Comments and Suggestions for Authors
This manuscript addresses a highly relevant and timely topic within the field of mental health and sport psychology, namely, the impact of social media use on muscle dysmorphia (MD) risk in young athletes. The integration of traditional psychometric methods with interpretable machine learning (ML) models is a notable strength that enhances the methodological rigor and innovation of the study.
The manuscript is well-organized, clearly written, and supported by a solid theoretical framework. Nonetheless, several aspects of the manuscript would benefit from clarification or further development, particularly regarding methodological transparency, theoretical grounding of some measurement choices, and discussion of limitations.
Strengths
- The use of both correlational methods and ML (CatBoost with SHAP values) adds depth and robustness to the analysis.
- The sample size is large and includes a variety of sport disciplines, increasing the ecological validity of findings.
- The discussion effectively integrates the study’s findings with existing literature and theoretical models (e.g., Objectification Theory, Tripartite Influence Model).
- The inclusion of validated instruments such as the MDDI-1 strengthens the psychometric foundation of the work.
Major Issues
- Operationalization of Social Media Variables:
The authors introduced a custom Social Behavior Questionnaire. While internal consistency is reported (Cronbach’s α = .76), further details are needed. Specifically:
- Was a factor structure tested (e.g., exploratory factor analysis)?
- Were item sources or theoretical justifications provided for the selected items?
- Please clarify how composite scores or constructs (if any) were derived.
- Statistical Rigor in Correlation Interpretation:
Correlation coefficients (ρ and τ) are interpreted as strong or robust, but caution is needed to avoid overstatement—especially given the cross-sectional nature and self-report data. It is recommended to more clearly distinguish statistical from practical significance. - Machine Learning: Reproducibility and Model Validation:
The implementation of CatBoost and SHAP is a strength; however, more details would improve transparency:
- Please report the RMSE and R² values for the model.
- Were features standardized or transformed before modeling?
- Was feature selection data-driven or based on theory?
- Include SHAP summary plots or provide more interpretation of feature interaction effects if available.
- Causal Claims:
While the authors appropriately acknowledge the cross-sectional design, certain phrasings (e.g., "predictors of MD symptoms") might suggest causality. Please revise these instances to clarify that associations—not causal relationships—are being examined. - Ethics Statement:
The ethical exemption rationale is clear and aligned with local requirements. However, it would strengthen the manuscript to specify whether the study was pre-registered or if a protocol is available on request.
Minor Suggestions
- Abstract: Consider replacing "almost perfectly aligned" (line 263) with a more precise term such as "strongly correlated" to maintain academic tone.
- Figure 3: Include statistical values (e.g., Kruskal–Wallis H and p-value) directly in the figure legend for clarity.
- Table/Figure captions: A few figure legends (e.g., Figures 2 and 3) could be expanded slightly to clarify sample size and variable measurement.
- Language: The manuscript is overall well-written, but minor grammar issues (e.g., subject–verb agreement, article use) could benefit from final proofreading.
This is a well-designed and methodologically innovative study that makes a valuable contribution to the literature on body image and social media. With some clarifications—especially around the construction of the custom questionnaire and further transparency in machine learning implementation—the manuscript would meet the high standards expected for publication in a peer-reviewed journal.
Round 3
Reviewer 1 Report
Comments and Suggestions for Authors
The authors have addressed all comments of the last round of revision.
Reviewer 3 Report
Comments and Suggestions for Authors
The authors have satisfactorily addressed and resolved all of my revisions.